# The Effects of 3-Dimensional Bioprinting Calcium Silicate Cement/Methacrylated Gelatin Scaffold on the Proliferation and Differentiation of Human Dental Pulp Stem Cells

**DOI:** 10.3390/ma15062170

**Published:** 2022-03-15

**Authors:** Dakyung Choi, Manfei Qiu, Yun-Chan Hwang, Won-Mann Oh, Jeong-Tae Koh, Chan Park, Bin-Na Lee

**Affiliations:** 1Department of Conservative Dentistry, School of Dentistry, Dental Science Research Institute, Chonnam National University, Gwangju 500-757, Korea; ekruddlspwlq@naver.com (D.C.); qiumanfei16@gmail.com (M.Q.); ychwang@jnu.ac.kr (Y.-C.H.); wmoh@jnu.ac.kr (W.-M.O.); 2Department of Pharmacology and Dental Therapeutics, Hard-Tissue Biointerface Research Center, School of Dentistry, Dental Science Research Institute, Chonnam National University, Gwangju 500-757, Korea; jtkoh@jnu.ac.kr; 3Department of Prosthodontics, School of Dentistry, Dental Science Research Institute, Chonnam National University, Gwangju 500-757, Korea

**Keywords:** bioprinting, gelatin, mineral trioxide aggregates (MTA), scaffold

## Abstract

A calcium silicate cement/methacrylated gelatin (GelMa) scaffold has been applied in tissue engineering; however, the research on its applications in dental tissue regeneration remains lacking. We investigate the effect of this scaffold on human dental pulp stem cells (hDPSCs). hDPSCs were cultured in 3D-printed GelMa and MTA-GelMa scaffolds. Cell adhesion was evaluated using scanning electron microscopy images. Cells were cultured in an osteogenic differentiation medium, which contained a complete medium or α-MEM containing aqueous extracts of the 3D-printd GelMa or MTA-GelMa scaffold with 2% FBS, 10 mM β-glycerophosphate, 50 μg/mL ascorbic acid, and 10 nM dexamethasone; cell viability and differentiation were shown by WST-1 assay, Alizarin Red S staining, and alkaline phosphatase staining. Quantitative real-time PCR was used to measure the mRNA expression of DSPP and DMP-1. One-way analysis of variance followed by Tukey’s post hoc test was used to determine statistically significant differences, identified at *p* < 0.05. hDPSCs adhered to both the 3D-printed GelMa and MTA-GelMa scaffolds. There was no statistically significant difference between the GelMa and MTA-GelMa groups and the control group in the cell viability test. Compared with the control group, the 3D-printed MTA-GelMa scaffold promoted the odontogenic differentiation of hDPSCs. The 3D-printed MTA-GelMa scaffold is suitable for the growth of hDPSCs, and the scaffold extracts can better promote odontoblastic differentiation.

## 1. Introduction

The regeneration of pulp and dentin has been developed using modern tissue engineering concepts and the discovery of dental stem cells. Gronthos et al. demonstrated the in vivo generation of dentin from pulp cells and observed pieces of human pulp/dentin complex formed ectopically in immune-compromised mice [1]. This finding was a start to further review the stem cell-based regeneration of pulp/dentin in clinical uses.

The therapeutic goal is the successful treatment of leading reasons through regenerating functional pulpal tissue, applying protocols related to regenerative endodontic procedures (REPs). REPs are described as “biologically based processes designed to replace damaged structures, such as dentin and root structures and the pulp-dentin complex cells” [2]. The strategy used in the present study is the local regeneration of the dentin–pulp complex by inducing the development of pulp cells, capillaries, and neurons through residual root pulp using a modified approach.

There are three important factors for tissue regeneration: cell, growth factor, and scaffold [3,4,5]. The induction of a capillary network and a closed space forming a suitable environment are also significant in the successful regeneration. The appropriate selection of the required growth factor, the delivery system of the growth factor, and a scaffold inducing stem cells and blood vessels from the residual pulp is essential to establish the local regeneration of the dentin–pulp complex.

The selection of a suitable scaffold material is crucial for cell-based tissue regeneration [6,7,8,9,10]. Gelatin is a mixture of peptides and proteins produced by the partial hydrolysis of collagen extracted from the skin, bones, and connective tissues of animals [11]. Gelatin is a major component of the extracellular matrix obtained from denatured collagen, and is a cell-adherent component [12] with good biodegradability and biocompatibility [13]. Due to these characteristics, it is considered an ideal material that could emulate the natural structure of the ECM [14]. However, it degrades quickly and has no long-term mechanical stability and its poor mechanical and thermal properties are its drawbacks as a biomaterial [15]. To overcome these challenges, gelatin has been modified with methacrylic anhydride. Methacrylated gelatin, named GelMa by van Den Bulcke et al. [15], is synthesized by binding the methacrylic groups on the gelatin surface through a covalent reaction [16]. Its properties are similar to those of gelatin and it exhibits high mechanical stability. It also has good bioactivity and allows cell attachment and proliferation in the GelMa scaffold [17,18,19].

Calcium silicate (CS)-based cements, such as mineral trioxide aggregate (MTA), are used in tissue regeneration because of its excellent sealing ability and biocompatibility [20,21]. MTA consists of tricalcium silicate, tricalcium aluminate, tetracalcium aluminoferrite, gypsum, bismuth oxide, and other mineral oxides that are reported to be less cytotoxic [22] than other endodontic materials and able to enhance the odonto/osteoblastic capacity of human dental pulp cells both in vivo and in vitro [23,24,25]. However, MTA has poor wear resistance and fracture toughness that limits its life span in a load-bearing environment [26].

Scaffolds designed from pulp regeneration therapy should mimic the microenvironment of the root canal and provide mechanical support; however, the use of traditional scaffolds has many problems, such as the restriction of the scaffold shape by the traditional process and poor reproducibility [27,28,29,30,31]. The 3D bioprinting method was developed as the newest method used to design a 3D framework for ideal structural scaffolds with a better control of cell proliferation and tissue formation. Three-dimensional printing can control scaffold morphology, pore size, and porosity better, and also allows the use of material without affecting its own properties [32,33,34,35]. Thus, the present study compared the effects of aqueous extracts of the GelMa or GelMa-MTA scaffolds made by a 3D bioprinter on the adhesion and differentiation of human dental pulp stem cells (hDPSCs). Therefore, the aim of this study is to investigate the effects of 3D-printed GelMa/MTA scaffold on cell viability and the odontogenic differentiation of hDPSCs.

## 2. Materials and Methods

### 2.1. Materials

Gelatin with methacryloyl groups (10 mL) was purchased from ROKIT healthcare (GelMa, INVIVO-GEL ESSENTIAL, ROKIT Healthcare, Seoul, Korea). A gelatin solution was prepared by warming the INVIVO-GEL Essential Syringe (ROKIT Healthcare) at 37 °C for approximately 30 min or until it became liquefied. To form bioink, Gel-linker (ROKIT Healthcare) was added to INVIVO-GEL Essential in a 1:4 volume ratio (1 mL Gel-linker to 4 mL INVIVO-GEL) and mixed well. The bioink was transferred to a dispensing container for use and maintained at 4 °C until the bioink solidified prior to use. Then, one of two types of MTA (ProRoot MTA; Dentsply Tulsa Dental, Tulsa, OK, USA; Endosem Zr; Maruchi, Wonju, Korea) powder (300 mg) was dropped into each of the bioinks. When the bioink hardened, it was dispensed for experiments and the temperature was maintained at room temperature or cooler.

### 2.2. Fabrication of 3D MTA-GelMa Scaffolds

A bio-dispenser in a Rokit INVIVO 3D printer (ROKIT Healthcare) was used to construct the GelMa scaffolds in a layer-by-layer deposition with the desired geometry. The Organ Regenerator Program (ROKIT Healthcare) was used to design the scaffold, which was saved as a gcode file. The gcode was then transferred to a printing machine for printing. The bioink was loaded into the dispensing container. The temperature of the bed was maintained at 4 °C. Subsequently, the material was extruded through a 0.2 mm nozzle at a printing speed of 10 mm/s under a constant pressure of 90 kPa; an 11 mm line with a height of 11 mm and a total of 18 lines were printed in parallel (Figure 1). The dispensed bioink was exposed to UV light (wavelength 365 nm, 5–10 min) until it reached the desired stiffness. Sterilization was carried out by soaking the scaffolds in 70% ethanol for 30 min. The 3D-printed scaffolds were lyophilized for 24 h at −70 °C.

### 2.3. Materials

The surface morphology of the scaffolds was examined using scanning electron microscopy (SEM; SNE-4500M, SEC, Suwon, Korea) with an accelerating voltage of 20 kV. The scaffolds were coated with gold for 120 s using a sputter coater. 

### 2.4. In Vitro Biocompatibility Evaluation

#### 2.4.1. Isolation and Culture of hDPSCs

Primary hDPSCs were purchased from Cell Engineering for Origin (CEFO CO. LTD., Seoul, Korea). hDPSCs were cultured in α-minimum essential medium (α-MEM; Gibco, Grand Island, NY, USA) supplemented with 10% fetal bovine serum (FBS; Gibco), 100 U/mL penicillin, and 100 mg/mL streptomycin (Gibco) in a humidified atmosphere of 5% CO_2_ at 37 °C. The medium was changed every 48 h. The cells were subcultured upon reaching confluence and were used at passages 6–7.

#### 2.4.2. Cell Adhesion on the 3D Scaffold

The hDPSCs were seeded at a density of 5 × 10^4^ cells per well on the scaffold in 24-well plates (Ultra-low Attachment Plate, Corning, Glendale, AZ, USA) and cultured for 3 days. The cells adhered to the scaffolds were fixed using 2.5% glutaraldehyde for 2 h at room temperature. The scaffold was washed twice with PBS, and subsequently dehydrated with serial dilutions of ethanol (40–100%) for 10 min each to preserve the cell morphology. The cell attachment and morphology of the scaffolds were examined using SEM with an accelerating voltage of 20 kV. The scaffolds were coated with gold for 120 s using a sputter coater.

#### 2.4.3. Material Extracts

The 3D-printed scaffolds of each test material were transferred into 50 mL tubes and incubated in growth media (α-MEM supplemented with 10% FBS and antibiotics) or induction media (2% FBS, antibiotics, 10 mM β-glycerophosphate, 50 μg/mL ascorbic acid, and 10 nM dexamethasone) for 24 h at 37 °C in a humidified atmosphere containing 5% CO_2_. These extracts were collected and then filtered using 0.20-μm filters (Minisart, Sartorius Stedim Biotech, Goettingen, Germany). The experimental group was treated with the material extract. The control group was treated with media only.

#### 2.4.4. Cell Viability

Cell viability was analyzed by the WST-1 assay using the EZ-Cytox assay kit (Dogenbio, Seoul, Korea) according to the manufacturer’s instructions. Briefly, the hDPSCs were seeded at a density of 5 × 10^4^ cells per well in 96-well plates and cultured at 37 °C for 24 h. The medium was gently sucked and rinsed twice with PBS then replaced with 100 μL of scaffold extract. After 24 h of hDPSC culture, 10 μL of Ez-Cytox was added to each well, followed by 1 h of incubation. After incubation, absorbance was measured at 420 nm using a microplate spectrophotometer (Multiskan GO Microplate Spectrophotometer; Thermo Scientific, Waltham, MA, USA) and software (SkanIt Software 4.1 for Microplate Readers; Thermo Fisher Scientific, Rockford, IL, USA). The control group was treated with growth media only. 

#### 2.4.5. Quantitative Real-Time Polymerase Chain Reaction (PCR)

The hDPSCs were seeded into six-well culture plates and pre-incubated in α-MEM containing 10% FBS and antibiotics for 3 days. The culture medium was replaced by material extracts for 2 days. Total RNA extraction from each group and complementary DNA synthesis were performed with TRIzol reagent (Invitrogen, Carlsbad, CA, USA) and the AccessQuick RT-PCR System (Pro-mega, Madison, WI, USA) according to the manufacturer’s instructions. Quantitative real-time PCR was conducted using a Quanti-Tect SYBR Green PCR Kit (Qiagen, Valencia, CA, USA) and all primers were synthesized by Bioneer (Daejeon, Korea). Thermal cycling conditions were as follows: 5 min at 95 °C followed by 45 cycles of 95 °C for 5 s, and 60 °C for 10 s. Relative gene expression was quantified after normalizing against the expression level of glyceraldehyde 3-phosphate dehydrogenase (GAPDH) as an endogenous control. The relative gene expression values were analyzed using the 2^(−ΔΔCt)^ method. The primer sequences used in this study are detailed in Table 1.

#### 2.4.6. Alkaline Phosphatase (ALP) and Alizarin Red S (ARS) Staining

hDPSCs were seeded at a density of 5 × 10^4^ cells per well on 24-well plates. Differentiation was induced by culturing hDPSCs in osteogenic differentiation medium, which contained complete medium or α-MEM containing aqueous extracts of the 3D-printed GelMa or MTA-GelMa scaffold with 2% FBS, 10 mM β-glycerophosphate, 50 μg/mL ascorbic acid, and 10 nM dexamethasone. Every 2 or 3 days, the osteogenic medium containing complete medium or aqueous extract medium was replaced. ALP activity was assessed using ALP staining after 7 days. Cells were washed with Dulbecco’s phosphate-buffered saline, fixed with 70% ethanol for 30 min, and rinsed with distilled water three times. Fixed hDPSCs were treated with 200 µL of ALP staining reagent (BCIP^®^/NBT Liquid Substrate System; Sigma–Aldrich, Rockford, IL, USA) per well. For quantitative evaluation, the stains were treated with 10% cetylpyridinium chloride (CPC, pH 7.0) for 30 min followed by absorbance measurement at a wavelength of 562 nm using a microplate spectrophotometer (Multiskan GO Microplate Spectrophotometer; Thermo Scientific, Waltham, MA, USA) and software (SkanIt Software 4.1 for Microplate Readers; Thermo Fisher Scientific, Rockford, IL, USA). 

The calcium deposition of the cultured dental pulp stem cells was assessed by ARS assay after 14 days. The medium was gently sucked and rinsed twice with PBS. Subsequently, the cells were fixed with 70% ethanol at 4 °C for 1 h and washed twice with deionized water to remove the fixative residues. ARS solution (2%) was added and incubated at room temperature for 30 min. Then, the staining solution was removed from each well and washed three times with deionized water. For quantity analysis, the ARS stain was extracted with 10% CPC. Finally, 100 μL of the solutions was transferred to a 96–well plate and their absorbance at 540 nm was measured using a microplate reader. 

### 2.5. Statistical Analysis

Each experiment containing triplicate independent samples was repeated at least twice and qualitatively identical results were obtained. One-way analysis of variance followed by Tuckey’s post hoc test was used to analyze all data using SPSS 25.0 software (Statistical Package for Social Science, version 25.0, 2020, IBM Corp, Armonk, NY, USA). A significant difference was identified at *p* < 0.05.

## 3. Results

### 3.1. Morphology of the Scaffolds

The morphology and microstructure of the scaffolds are shown in Figure 2. The SEM images of all scaffolds showed a dense and uniform microstructure of the surface on both the GelMa and MTA-GelMa scaffolds. The wall of the GelMa scaffold had a smooth surface (Figure 2A,B), whereas the MTA-GelMa scaffolds exhibited MTA particles introduced into the surfaces of the walls (Figure 2C–F). The incorporation of a small amount of MTA did not affect the whole structure of the scaffold.

### 3.2. Cell Adhesion 

To understand the cell distribution and bioactivity of the scaffolds, hDPSCs were seeded onto the 3D-printed GelMa or MTA-GelMa scaffolds and SEM analysis was conducted after 3 days of culture. From the SEM observation, it was evident that hDPSCs were distributed evenly and attached to the surfaces and inner sides of all scaffolds (Figure 3). 

### 3.3. Cell Viability

The viability of the cells was first measured by culturing hDPSCs in the extracted solutions of the scaffolds for 24 h. Compared with the control, there were no significant differences in OD values of the GelMa and MTA-GelMa scaffold groups (*p* > 0.05). Therefore, neither GelMa nor MTA affected cell viability (Figure 4).

### 3.4. Odontogenic Differentiation and Mineralization

The messenger RNA expression levels of DSPP and DMP-1 were detected using real-time reverse transcription PCR (Figure 5). The mRNA expressions of DSPP were significantly higher in the GelMa/Proroot MTA groups compared with the GM and IM groups after 3 days (*p* < 0.05). The mRNA expression levels of DMP-1 were higher in the GelMa/ProRoot MTA scaffold compared with GM and IM after 3 days, but with no significance (*p* > 0.05). After 5 days, a higher expression level of DMP-1 was observed in the MTA-GelMa groups than in the GM and IM groups, although there were no significant differences (*p* > 0.05). These results support a role for 3D-printed MTA-GelMa scaffold extracts in the odontoblastic differentiation of hDPSCs. 

The ALP assay was used to estimate the cell differentiation capability of the scaffolds, and the ALP activity after 7 days is depicted in Figure 6. The results indicate that the ALP staining of cells after culture in the presence of either 3D-printed GelMa or MTA-GelMa scaffold aqueous extracts were significantly higher (*p* < 0.05) than those of the GM and IM groups. 

ARS was used to determine the amount of calcium deposit in the scaffolds. Figure 7A shows the optical images for the ARS staining of the hDPSCs cultured on the scaffolds for 14 days, and Figure 7B shows the optical densities of ARS extracted from stained cultures to quantify the calcium deposits. The results indicate that the optical density of the GelMa and GelMa/ProRoot MTA scaffolds was higher than that of the IM group, but with no significance (*p* > 0.05). However, the MTA/Endocem Zr scaffold exhibited significantly higher calcium deposition than the control group (*p* < 0.05). These results indicate that the MTA-GelMa scaffold might stimulate mineralized nodule formation and calcium deposition. 

## 4. Discussion

For the success of dentin/pulp regeneration, substantial efforts have been made to develop suitable biomaterials and ideal structures for scaffolds. Currently, biomaterials play a crucial role in promoting cell responses, such as attachment, proliferation, and differentiation [26,36,37,38]. Furthermore, the porosity, pore size, and pore interconnectivity contribute toward the tissue growth and diffusion of nutrients [39,40]. Three-dimensional printing has been introduced as a new method for making scaffolds; we therefore applied 3D printing technology and designed 3D scaffolds to fabricate appropriate architecture for tissue engineering [38,41,42,43].

First, we evaluated the biocompatibility of the scaffolds by comparing the effects of these materials on cell viability. In this study, SEM images showed that hDPSCs cultured on scaffolds appeared well attached; this was true for every scaffold investigated. Furthermore, ProRoot MTA and Endocem Zr had similar effects on cell viability in the WST-1 assay and GelMa also had no cytotoxicity. These results are consistent with those of a previous study, wherein no significant differences were shown between ProRoot MTA and Endocem Zr in a cell viability assay [44,45].

Next, we evaluated whether the extraction of ProRoot MTA and Endocem Zr scaffolds facilitated the odontoblastic differentiation of hDPSCs. ProRoot MTA consisted of 75% Portland cement, 5% calcium sulfate dehydrate, and 20% bismuth oxide [46]. Unlike ProRoot MTA, in which Ca and Si are predominant, Endocem Zr consists of 27–37% calcium oxide, 7–11% silicon dioxide, 3–5% aluminum oxide, 1.7–2.5% magnesium oxide, 1.3–2.3% ferrous oxide, and 43–46% zirconium oxide, which is composed of Ca and Si [44,47,48]. In Endocem Zr, zirconium oxide has been employed as a radiopacifier; the substitution of bismuth oxide with zirconium oxide contributes to setting the reaction time. Lee et al. previously reported that Endocem Zr released the lowest amount of Ca ions among the CS-based cements in ARS staining [44]. Another study also reported that compared with White MTA, Endocem MTA and Endocem Zr are associated with significantly less Ca ion release and produce apatite-like crystalline precipitates of significantly lower Ca/P ratio when immersed in PBS [49].

However, in the present study, Endocem Zr showed significantly more calcium deposits compared with the control group in ARS staining. This was consistent with the results of a previous study by Chung et al., who found that ProRoot MTA, Endocem Zr, and Retro MTA showed significant increases in calcium concentration, especially in Endocem Zr [47].

We then investigated whether the tested materials facilitate the odontoblastic differentiation of hDPSCs, as evidenced by the expression of odontogenic-related markers, such as DSPP and DMP-1. The mRNA expressions of DSPP were significantly higher in GelMa/ProRoot MTA groups compared with the GM and IM groups after 3 days. The mRNA expressions of DMP-1 were higher in the MTA-GelMa groups compared with the GM and IM groups after 5 days, but with no significance.

Ca ions are released from ProRoot MTA in phosphate-containing environments, and they induce the development of a superficial layer of apatite on the cement surface [50]. However, the controversial issue of the differentiation potential variation between studies could be caused by differences in experimental conditions. One of the most influential factors may be the concentration of the material extracts used in the experiment. In the present study, we used 300 mg of MTA per scaffold. Higher extract concentrations can lead to a higher toxic content and pH, which can lead to poor test results [51].

Our results provide evidence that MTA-GelMa scaffolds can increase mineralized nodule formation and may induce odontogenic differentiation. We believe that these novel 3D-printed MTA-GelMa scaffolds could prove to be innovative bioscaffolds for the development of dental and bone regeneration.

However, there are limitations to this study. We used the aqueous extracts of the scaffolds instead of the scaffolds directly in the present study. For the clinical use of MTA-GelMa scaffolds, it is thought that scaffold should be able to be applied directly to the cell rather than the extract. In addition, there is a need to see odontoblastic marker expression in protein level as well as RNA level, and furthermore, the biologic mechanism for this will also be confirmed.

## 5. Conclusions

The present study evaluated the effect of a 3D-printed MTA-GelMa scaffold on hDPSCs. hDPSCs adhered well to both the 3D-printed GelMa and MTA-GelMa scaffolds. GelMa and MTA-GelMa groups showed good biocompatibility in the cell viability test. The 3D-printed MTA-GelMa scaffold promoted the odontogenic differentiation of hDPSCs compared with control. Therefore, the 3D-printed MTA-GelMa scaffold is biocompatible for the growth of hDPSCs, and the scaffold extracts promote odontoblastic differentiation better. We suggest that the 3D printed MTA-GelMa scaffold may be a potential bioscaffold for dentin/pulp tissue regeneration.

## Figures and Tables

**Figure 1 materials-15-02170-f001:**
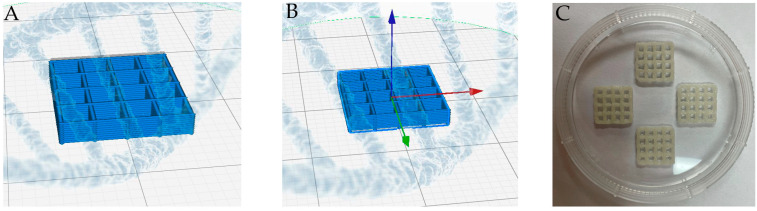
(**A**,**B**) Design of the 3D scaffold using the Organ Regenerator Program. (**C**) A photo image of the 3D scaffolds is a figure.

**Figure 2 materials-15-02170-f002:**
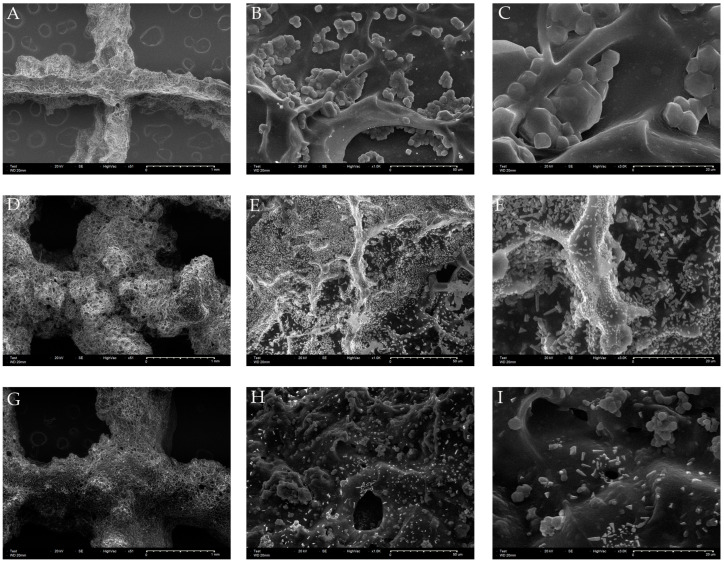
(**A**–**C**) SEM images of the 3D-printed GelMa scaffolds. (**D**–**F**) SEM images of the 3D-printed GelMa/ProRoot MTA scaffolds. (**G**–**I**) SEM images of the 3D-printed GelMa/Endocem Zr scaffolds. (**A**,**D**,**G**) ×50 magnification, (**B**,**E**,**H**) ×1000 magnification, (**C**,**F**,**I**) ×3000 magnification.

**Figure 3 materials-15-02170-f003:**
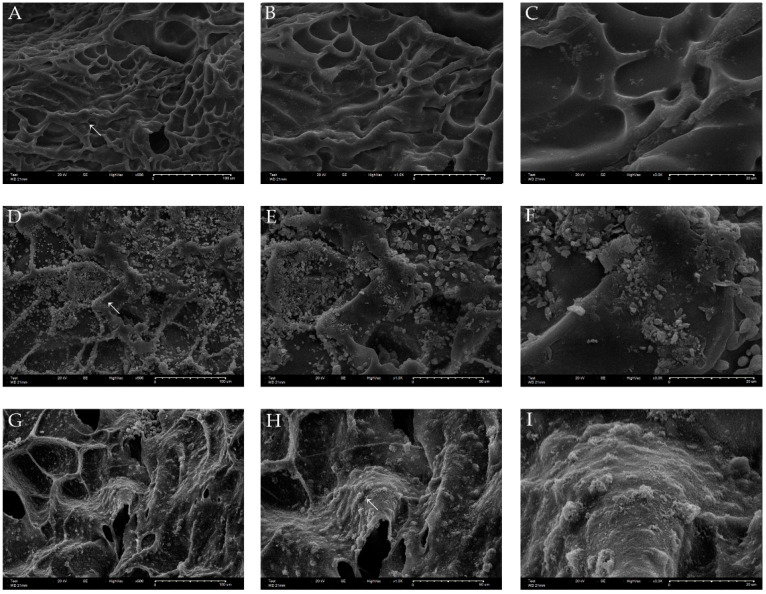
(**A**–**C**) Adhesion of hDPSCs on the 3D-printed GelMa scaffolds. (**D**–**F**) Adhesion of hDPSCs on the 3D-printed GelMa/ProRoot MTA scaffolds. (**G**–**I**) Adhesion of hDPSCs on the 3D-printed GelMa/Endocem Zr scaffolds. (**A**,**D**,**G**) ×500 magnification, (**B**,**E**,**H**) ×1000 magnification, (**C**,**F**,**I**) ×3000 magnification.

**Figure 4 materials-15-02170-f004:**
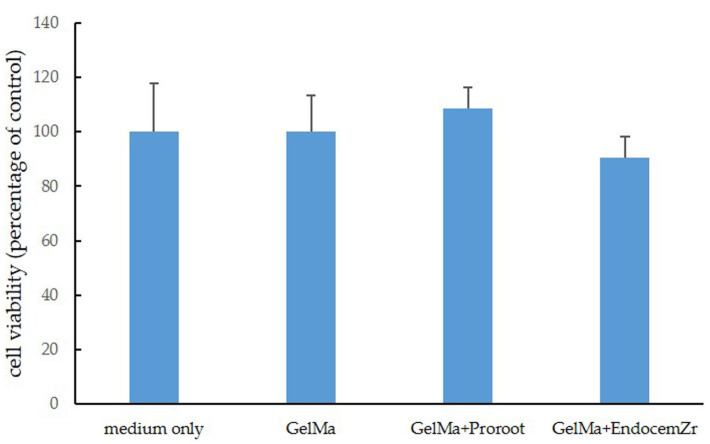
The cell viability of hDPSCs exposed for 24 h to extracts of the test materials as measured by the WST-1 assay. There was no significant difference in the cell viability of the GelMa and MTA-GelMa scaffold groups compared to the control (*p* > 0.05).

**Figure 5 materials-15-02170-f005:**
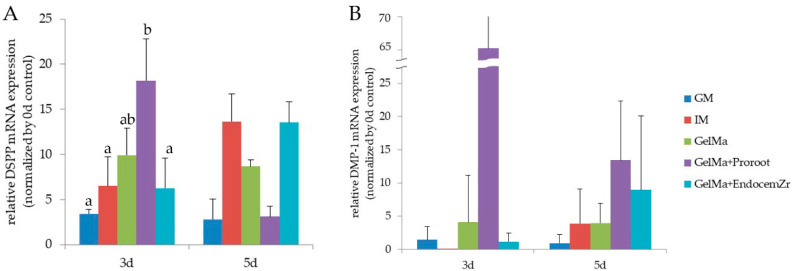
The mRNA expression level of odontogenic markers was determined by quantitative real-time PCR. The expression of (**A**) DSPP and (**B**) DMP-1 in hDPSCs stimulated with the extraction of scaffolds was determined. Different lower-case letters represent statistically significant differences (*p* < 0.05).

**Figure 6 materials-15-02170-f006:**
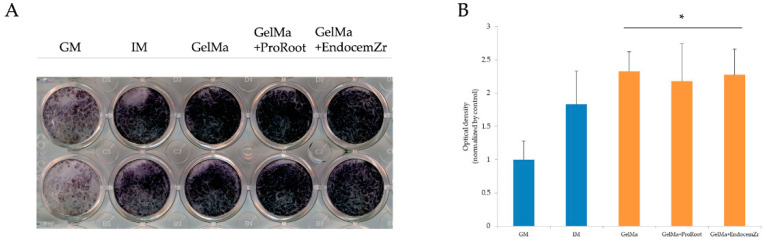
(**A**) ALP activity of hDPSCs cultured on the extracts of scaffolds for 7 days. (**B**) ALP staining of hDPSCs cultured on the extracts of scaffolds for 7 days. * Statistically significant difference (*p* < 0.05) compared to the control group.

**Figure 7 materials-15-02170-f007:**
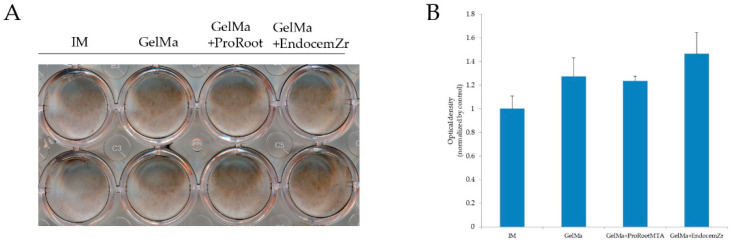
(**A**) Quantification of calcium mineral deposits of hDPSCs cultured on the extracts of scaffolds for 14 days. (**B**) Alizarin Red S staining of hDPSCs cultured on the extracts of scaffolds for 14 days. There was no significant difference in the calcium deposition of the GelMa and MTA-GelMa scaffold groups compared to the control (*p* > 0.05).

**Table 1 materials-15-02170-t001:** Primer sequences used for real-time PCR.

	Primer Sequences (5′–3′)
*dspp*	Forward: TTAAATGCCAGTGGAACCAT Reverse: ATTCCCTTCTCCCTTGTGAC
*dmp-1*	Forward: TGGGGATTATCCTGTGCTCT Reverse: TACTTCTGGGGTCACTGTCG
*gapdh*	Forward: AAGGTGAAGGTCGGACTCAAC Reverse: GGGGTCATTGATGGCAACAATA

## Data Availability

Data sharing is not applicable for this paper.

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
