# Peer review of "The Effects of 3-Dimensional Bioprinting Calcium Silicate Cement/Methacrylated Gelatin Scaffold on the Proliferation and Differentiation of Human Dental Pulp Stem Cells"

_materials, 2022, doi:10.3390/ma15062170_

Round 1

Reviewer 1 Report

Dear authors,

The proposed article highlights the issues and potential better approaches for the formulations and fabrication of 3D scaffolds for bone regeneration.

However, similar solutions to the one proposed by the authors are already reported in the literature. Innovative aspects of the presented research should be better described considering the huge number of papers published in this field.

Introduction - comprises of a presentation of the chosen components for the scaffold, rather than presenting the state-of-the-art and setting out the rationale behind the design of this work. An introduction should be a cohesive section to describe the topic (which is adequately described in the first line/sentence), followed by a brief review of the most recent bibliography, the rationale, the description of the actual work with a focus on its novelty and significance and finally the outline of the paper’s contents. In my opinion, it should be revised accordingly.

L185 -please give a name to the subsection

Figures 2 and 3 are not representative of a 3D printed scaffold, the authors should also offer a smaller magnification/optical microscopy image. Their resolution is terrible. I don’t see the arrow pointing to the hDPSCs adhered on the surface (as stated in the figure’s name)

Figure 4: please insert the statistical evaluation results, please change the axes, and upload a higher resolution image.

The Discussion section seems likely to be only a Conclusion.  Please give an extended discussion on the obtained results and correlate your findings with previous literature studies and prospective applications. 

The Conclusion section must be properly written.

The reference list is rather small and outdated.

Best regards

Reviewer 2 Report

This research is under the scope of this journal; the topic is relevant for readers, and this research deals with potentially significant knowledge of the field. 

Introduction

 - Abstract gives information on the main feature of the performed study, but some details about the experimental practice must be added.

- Authors must clarify the necessity of the performed research. The objectives of the study must be clearly mentioned in the introduction.

- The literature study must be enriched. In this respect, authors must read and refer to the following papers: (a) https://doi.org/10.3390/biomedicines9010024 CS have good biocompatibility. (b) https://doi.org/10.1007/s00784-021-03840-9. However, the introduction is too short in the current version.

Materials and Methods

  • When mentioning materials or devices: for some of them, you don't mention the manufacturer at all, for some you mention only the manufacturer, for some the manufacturer and city, for some you mention the manufacturer and city/ country. Standardized this presentation.
  • How many observers did the scoring? What is the interobserver agreement? How many times was performed done the determinations?
    • Which data are parametric and non-parametric? Please specify?
    How was the sample calculated? Did the authors perform a power analysis to evaluate if this sample size for the group was appropriate? How many samples are for the group? And How many samples for an animal? Please, add this information experimental design. 

Results

- Improve the resolution quality of all figures and graphs (and the size of the pictures). The font/language in the figure/caption is different from the text. Please, standardized the size and the font in the figures and charts with the font of the manuscript

- Identify the arrow

Discussion 

  • Please, clarified what was the limitation of this study?
  • Suggestion, the authors should finish the discussion suggesting further studies to evaluate the approach scaffold (clarified the future perspectives).

References

  • But references are not standardized. The titles of references have a different format, the title of the article is written in capital letters at the beginning of words, others only in lower case.

Round 2

Reviewer 1 Report

Dear authors,

I appreciate your efforts in modifying the manuscript and increasing its scientific value, by tackling most of my previous comments. However, some aspects still need to be taken into consideration before approval:

  1. The Conclusion Section needs major revision. It is unusual to have only one phrase.
  2. The Reference list is still outdated...there are only 15-17% sources from 2020-2022. 

Author Response

  1. The Conclusion Section needs major revision. It is unusual to have only one phrase. -> Thank you for your comment. We've discussed and concluded it enough in discussion section. And as you mentioned, we've supplemented the conclusions.
  2. The Reference list is still outdated...there are only 15-17% sources from 2020-2022. -> The reference has been updated to the latest.  Thank you.

Reviewer 2 Report

This research is under the scope of this journal; the topic is interesting for readers and this research deals with potentially significant knowledge to the field and an open new way for future studies.

The authors improved the quality of the manuscript after the reviewer's indications.

Author Response

Thank you.

This manuscript is a resubmission of an earlier submission. The following is a list of the peer review reports and author responses from that submission.